# CRISPR/Cas9-Mediated Knockout of the *HuR* Gene in U251 Cell Inhibits *Japanese Encephalitis* Virus Replication

**DOI:** 10.3390/microorganisms12020314

**Published:** 2024-02-02

**Authors:** Sai-Qi Luo, San-Jie Cao, Qin Zhao

**Affiliations:** 1Research Center for Swine Diseases, College of Veterinary Medicine, Sichuan Agricultural University, Chengdu 611130, China; 2021203028@stu.sicau.edu.cn; 2Sichuan Science-Observation Experimental Station of Veterinary Drugs and Veterinary Diagnostic Technique, Ministry of Agriculture and Rural Affairs, Chengdu 611130, China; 3National Demonstration Center for Experimental Animal Education, Sichuan Agricultural University, Chengdu 611130, China

**Keywords:** CRISPR/Cas9, gene knockout, HuR, U251 cell lines, JEV

## Abstract

Human antigen R (HuR) is an RNA-binding protein that regulates the post-transcriptional reaction of its target mRNAs. HuR is a critical factor in cancer development and has been identified as a potential target in many cancer models. It participates in the viral life cycle by binding to viral RNAs. In prior work, we used CRISPR/Cas9 screening to identify HuR as a prospective host factor facilitating Japanese encephalitis virus (JEV) infection. The *HuR* gene was successfully knocked out in U251 cell lines using the CRISPR/Cas9 gene-editing system, with no significant difference in cell growth between U251-WT and U251-HuR-KO2 cells. Here, we experimentally demonstrate for the first time that the knockout of the *HuR* gene inhibits the replication ability of JEV in U251 cell lines. These results play an essential role in regulating the replication level of JEV and providing new insights into virus–host interactions and potential antiviral strategies. It also offers a platform for investigating the function of HuR in the life cycle of flaviviruses.

## 1. Introduction

Human antigen R (HuR), or ELAVL1 as it is also named, is a crucial RNA-binding protein member of the embryonic lethal abnormal vision (ELAV) family. HuR is highly expressed in various tissues, such as brain, reproductive organs, fat, intestine, spleen, and testis [1]. HuR has been found to play crucial roles as a driver and promoter of various cancers [2,3,4], and is also identified as a key target in many cancer models, influencing tumor development primarily by regulating tumor cell invasion, migration, and proliferation [5,6]. HuR can also bind with the adenine–uracil-rich element (ARE), and ARE is located in the 3′ untranslated region (UTR) of target mRNAs. Further research has shown that HuR also plays a role in the life cycle of viruses by binding with viral genomic RNA during viral infection [7,8]. Investigating the interaction between HuR and diverse viral pathogens is significant. The few prior studies on the correlation between HuR and viruses focus primarily on the RNA viruses, especially on the family Flaviviridae, finding that HuR interacts with human immunodeficiency virus (HIV) type 1 reverse transcriptase and modulates reverse transcription in infected cells [9]. Sindbis virus (*Togaviridae* family) causes a dramatic relocalization of the cellular HuR protein from the nucleus to the cytoplasm in infected cells [10], usurps the HuR protein to avoid the cellular mRNA decay machinery, and maintains a highly productive infection [11]. HuR promoted EV71 (enterovirus 71) (*Picornaviridae* family) internal ribosomal entry site activity and virus replication [12]. HuR enhances Hepatitis C virus (HCV) (*Flaviviridae* family) replication by promoting the recruitment of replication machinery to viral 3′ UTR [13]. HuR-regulated apoptosis is induced by the encephalomyocarditis infection or the Semliki Forest virus (SFV) (*Flaviviridae* family) [14]. The Classical swine fever virus’ (CSFV) (*Flaviviridae* family) 3′ UTR binding proteins were what HuR was specifically binding to in the ARE region [15]. HuR was found to modulate Zika virus (ZIKV) (*Flaviviridae* family) replication [16].

These studies suggest there is a molecular mechanism involved in virus–host interaction, which probably has a significant impact on virus replication, cytopathology, and pathogenesis. And the impact of RNA viruses on the post-transcriptional regulation of cellular gene expression is unclear. Japanese encephalitis virus (JEV) is not only an important member of the *Flavivirus* genus family *Flaviviridae*, but is also a mosquito-borne, zoonotic flavivirus causing viral encephalitis in humans and reproductive disorder in swine. However, the role of host HuR in JEV replication has not yet been explored.

CRISPR/Cas9 has rapidly become the leading genome editing technology due to its precision, efficiency, and versatility. In agricultural and veterinary sciences, CRISPR-Cas9 is now widely used in applications like producing healthy animals with economically favorable traits through CRISPR/Cas9 gene editing, modifying viral genomes with CRISPR-Cas9, developing viral vector vaccines enabled by CRISPR/Cas9, and utilizing CRISPR/Cas9 systems to study virus–host cell interactions [17].

The U251 cell lines are a well-established line of human glioblastoma cells. Additionally, U251 cell lines are highly susceptible to various viruses, making them an excellent model for studying viral infections [18,19]. The U251 cell lines are highly susceptible to diseases caused by various flaviviruses. This susceptibility allows us to explore the entire viral life cycle, including viral entry, replication, assembly, and release within these cells. By leveraging U251 cell lines as an in vitro model, we can gain insights into the molecular mechanisms of flavivirus infections and the host–virus interactions that occur throughout the infection process [20,21,22,23,24].

In our preliminary study, genome-wide CRISPR/Cas9 screens have identified that host factor HuR might affect the ability of JEV to infect host cells. To further study the function and role of HuR in JEV-infected cells, in this study, HuR knockout U251 cells were successfully constructed using the CRISPR/Cas9 gene-editing system. And we conducted the first study examining the impact of HuR on JEV infection. This will provide an experimental model for further investigation into the specific role of HuR, in the invasion of nerve cells by *flaviviruses*.

## 2. Materials and Methods

### 2.1. Cell Culture and Virus

Human embryonic kidney (HEK)-293T cell lines (ATCC no.: CRL-3216) were purchased from the American Type Culture Collection (ATCC, Manassas, VA, USA). Professor Chao Huang from Sichuan Agricultural University generously provided the U251 cell lines. All cells were cultured in Dulbecco’s modified Eagle medium (DMEM, Gibco, Grand Island, NY, USA) and supplemented with 10% fetal bovine serum (Gibco, Invitrogen, Carlsbad, CA, USA), 100 U/mL penicillin, and 0.1 mg/mL streptomycin, at 37 °C in a 5% CO_2_ humidified incubator. The JEV strain (SA14) was used and maintained in our lab, and the viral titer was 5.0 × 10^7.0^ PFU/mL, GenBank accession MK585066.1.

### 2.2. Selection and Design of sgRNA

Referring to the human *HuR* genome sequence published by the National Center for Biotechnology (NCBI) database (ID: 1994), HuR sgRNA was designed using the CRISPR design tool on the website (http://chopchop.cbu.uib.no/ (accessed on 24 November 2022)). The fourth-ranked sgRNA targeting exon regions was selected. CACC nucleotide and AAAC nucleotide overhangs were added to the 5′ and 3′ ends of the sense and antisense strands, respectively, to enable cloning. The sgRNA oligonucleotide (Listed in Table 1) chain was synthesized by the Shanghai Sangon Bioengineering Company (Shanghai, China).

### 2.3. Construction of LentiCRISPR-V2 Recombinant Plasmid

The sgRNA upstream and downstream primers were diluted to 100 μM working stocks and annealed utilizing a standardized protocol to generate double-stranded DNA fragments. The LentiCRISPR-V2 plasmid backbone contains an ampicillin-resistance gene, which is suitable for ampicillin selection. The LentiCRISPR-V2 plasmid was linearized by restriction digestion with BsmBI. The prepared linearized vector and sgRNA-containing double-stranded DNA fragments were then ligated using T4 DNA ligase to construct the complete sgRNA-expressing lentiviral plasmid. The ligated products were transformed into ampicillin-resistant cells of the *Escherichia coli* strain DH5α and plated on LB agar plates containing l00 μg/mL ampicillin for clone selection. Then, the positive clones were sequenced by the Shanghai Sangon Bioengineering Company (Shanghai, China).

### 2.4. Lentivirus Packaging and Infection

HEK-293T cells were seeded into 6-well plates. When the cells reached approximately 85% confluence, the recombinant plasmid LentiCRISPR-V2 was packaged into lentivirus in the HEK-293T cells by co-transfection with psPAX2 and pMD2.G at a 5:3:2 ratio using the Lipofectamine^®^ 3000 Transfection Kit (Invitrogen, Carlsbad, CA, USA) and following the manufacturer’s instructions. The lentivirus cell-containing supernatant was collected after being transfected for 48 h by the recombinant plasmid LentiCRISPR-V2, psPAX, and pMD2.G, and centrifuged at 5000 rpm for 10 min at 4 °C, and then the supernatant was passed through 0.45 µm filters before use. When the confluence of U251 cell lines reached 40%, lentivirus was used to infect the cells. Then, Polybrene was added at a final concentration of 8 μg/mL to increase the efficiency of viral infection. At 36 h post-infection, the medium was replaced, and puromycin (Beyotime Biotechnology, Shanghai, China) was added at a final concentration of 5 μg/mL for selection.

### 2.5. Acquisition of Monoclonal Cells

*HuR* gene knockout U251 cell lines preliminarily selected via puromycin treatment were seeded by limited dilution into 96-well plates to allow monoclonal expansion from single cells. Genomic DNA was extracted from the cultured cells using a commercial kit (TIANGEN, Beijing, China) and utilized as the template for PCR analysis. The amplified PCR products were then submitted to the Shanghai Sangon Bioengineering Company (Shanghai, China) for Sanger sequencing to confirm editing outcomes.

### 2.6. Identification of HuR Gene Knockout U251 Cell Lines

Cell suspensions of U251-WT, U251-HuR-KO1, and U251-HuR-KO2 were harvested from 6-well cell culture plates. After centrifugation, the total cellular protein was extracted by lysing the cell pellets in RIPA buffer. Equal amounts of extracted proteins were separated by 12.5% SDS-PAGE and subsequently transferred to PVDF membranes via wet electroblotting. The membranes were blocked with 5% non-fat milk in TBST buffer for 1 h at room temperature to prevent nonspecific antibody binding. The membranes were then probed with specific primary antibodies targeting HuR and GAPDH proteins and followed by HRP-conjugated goat anti-rabbit IgG secondary antibody incubation. Protein bands were visualized using enhanced chemiluminescence (ECL) substrate and imaging.

### 2.7. Counting Kit-8 (CCK-8) Assay

Cell viability was determined using the CCK-8 assay (Beyotime Biotechnology, Shanghai, China). U251-WT and U251-HUR-KO2 cells were seeded into a 96-well cell culture plate. At 12 h and 24 h of culture, 10 µL of CCK-8 reagent was added to each well and incubated at 37 °C for 1 h. The absorbance of each well was measured using a microplate reader at 450 nm.

### 2.8. Quantitative Real-Time PCR (qRT-PCR)

The JEV *E* gene encodes the viral envelope protein (E protein) [25], and the E protein is the most important viral structural protein in JEV and most yellow viruses. It brings the virus into the cell in fusion with the cell membrane in association with cell membrane receptors and can induce the production of neutralizing antibodies [26,27]. Hence, the expression level of the *E* gene can serve as a reference for the replication and infection of the JEV. To assess the impact of *HuR* gene knockout on JEV replication, U251-WT and U251-HuR-KO2 cell lines were infected with JEV at an MOI of 0.1. Cell lysates were collected 12 h and 24 h post-infection, and the copy number of the JEV E gene in both cell types was determined using RT-qPCR. Cell samples were collected from 24-well cell culture plates, and RNA was extracted using the Multisource Total RNA Miniprep Kit (AxyPrep, Union City, CA, USA). For qPCR, reverse transcription was performed using the TransScript^®^ All-in-One First-Strand cDNA Synthesis SuperMix (TransGen, Beijing, China). The expression of the JEV E gene in cell lines was quantified by the PerfectStart^®^ Green qPCR SuperMix (TransGen, Beijing, China); GAPDH was used as an internal control. Real-time PCR was performed on the CFX Connect™ Real-Time PCR Detection System and analyzed using CFX Manager (Bio-Rad, Hercules, CA, USA). The primer sequences were as shown in Table 2.

### 2.9. Statistical Analysis

GraphPad Prism (version 9.5; GraphPad Software, Inc., La Jolla, CA, USA) was used for statistical analysis. Statistical data from three separate experiments were presented as mean ± standard deviation. Statistical comparisons between the two groups were performed using unpaired *t*-tests. *p* < 0.05 was considered to indicate a statistically significant difference. *p* < 0.05, 0.01, and 0.001 were indicated by asterisks *, **, and ***, respectively. ns means *p* > 0.005.

## 3. Results

### 3.1. Construction of the LentiCRISPR V2-sgRNA Recombinant Plasmid

The designed sgRNA is shown (Figure 1A), which comes from the fourth exon of the *HuR* gene sequence. The double-stranded sgRNA was ligated to the linear LentiCRISPR V2 plasmid digested by the BsmB I enzyme (Figure 1B). The ligation products were transformed into DH5α competent cells. Clones were then verified using colony PCR with appropriate primers. And the positive clones were selected by agarose gel electrophoresis (Figure 1C). Sequencing results showed that the recombinant plasmid was correctly constructed (Figure 1D). The constructed recombinant plasmids were named LentiCRISPR V2-HuR-sgRNA1 and LentiCRISPR V2-HuR-sgRNA2.

### 3.2. Screening and Sequencing of HuR Gene Knockout Monoclonal Cells

LentiCRISPR V2-HuR-sgRNA1 and LentiCRISPR V2-HuR-sgRNA2 were co-transfected into HEK-293T cells and lentivirus packing helper plasmids using liposome transfection reagents. The lentivirus was collected and injected into U251 cell lines. After puromycin screening, the cells were subjected to a limited dilution method in 96-well plates to obtain monoclonal cells. Genomic DNA extracted from monoclonal lines was subjected to PCR amplification and Sanger sequencing. This revealed one nucleotide deletion in exon two of the *HuR* gene encoding HuR in the U251-HuR-KO2 clone compared to wild-type parental cells, indicating successful CRISPR/Cas9-mediated *HuR* gene knockout. In contrast, no editing was detected in the U251-HuR-KO1 clone (Figure 1E).

### 3.3. Western Blot Analysis of HuR Gene Knockout Monoclonal Cells

Cells were collected from 6-well plates (with 8 × 10^5^ cells per well), and total proteins were extracted after RIPA cell lysis for the Western blot experiment, as shown in the Western blot image (Figure 1F). Western blot analysis revealed decreased HuR protein levels in the U251-HuR-KO1 clone compared to U251-WT cell lines. In contrast, HuR protein expression was completely ablated in the U251-HuR-KO2 clone. Combined with the genomic sequencing results, the U251-HuR-KO2 clone was selected for subsequent experiments as it represented a complete *HuR* gene knockout. The Western blot confirmed the lack of HuR protein expression in U251-HuR-KO2 cells. In summary, these data demonstrate successful CRISPR/Cas9-mediated generation of *HuR* gene knockout U251 cell lines, which will serve as a valuable tool for elucidating HuR functions.

### 3.4. Effect of HuR Gene Knockout on Cell Viability in U251 Cell Lines

Results showed that *HuR* gene knockout did not affect U251 cell line viability (Figure 2A). The morphology of two types of cells was examined under a light microscope. There was no significant difference in cell morphology (Figure 2B). Based on these findings, we will conduct further studies on viral infection in U251-HuR-KO2 cells.

### 3.5. Effect of Knockout of HuR Gene on JEV Replication

To characterize the functional effects of *HuR* gene knockout in U251 cells, JEV was utilized to infect the CRISPR-generated *HuR* gene knockout U251 lines. U251-WT and U251-HuR-KO2 cell lines were infected with JEV at a MOI of 0.1 and supernatants from infected cells were collected at 6, 12, 24, 36, and 48 hpi. The JEV *E* gene Ct values’ analysis uncovered suppressed JEV replication in U251-HuR-KO2 cell lines relative to U251-WT (Figure 3A). As depicted in Figure 3B, we used qRT-PCR analysis for U251-WT and U251-HuR-KO2 cells infected with JEV for 12 h and 24 h respectively. The JEV *E* gene Ct values in U251-WT cells were significantly reduced compared to U251-HuR-KO2 cells. These analyses revealed that the knockout of *HuR* genes significantly inhibited JEV replication. Western blot analysis of protein samples from U251-WT and U251-HuR-KO2 cells infected with JEV for 12 h and 24 h revealed similar results (Figure 3C), with a significant decrease in JEV E protein expression in U251-HuR-KO2 cells compared to U251-WT cells. In summary, the knockout of HuR inhibits the replication of JEV in U251 cell lines.

## 4. Discussion

As we know, host factors together with viral proteins regulate virus RNA replication. Hence, knockdown or knockout of related host genes has been essential in studying the interaction between host cells and the virus. Based on our preliminary study, we have identified that host factor HuR might affect the ability of JEV to infect host cells by using a genome-wide CRISPR screen. In this study, we performed the CRISPR/Cas9 gene-editing system to successfully construct *HuR* gene knockout U251 cell lines. The results of genome sequencing, i.e., Western blot, showed the U251-HuR-KO2 clone was better than U251-HuR-KO1 clone, so we selected the U251-HuR-KO2 clone as the cell model for the next study. Here, the cell viability and the cell morphology determined that there was no significant difference between the cell viability of U251-WT and U251-HuR-KO2 cells. Then, we used U251-WT and U251-HuR-KO2 cells to examine the impact of HuR on JEV SA14 strain infection after 12 h and 24 h. When U251-WT and U251-HuR-KO2 cells were infected with JEV (MOI = 0.1) for 12 h and 24 h, the qRT-PCR and WB analysis showed that the knockdown of HuR significantly reduced the JEV *E* gene mRNA levels’ copy number and E protein levels. Our results demonstrate for the first time the importance of host factor HuR for JEV replication.

Studying host factors using knockout target genes has become a well-recognized practical approach to investigating their functions. The emergence of the CRISPR/Cas9 system has demonstrated its enormous potential in gene editing [28]. The CRISPR/Cas9 system has rapidly gained popularity as a powerful tool for editing a wide range of genomes due to its ease of design, simplicity of use, and efficiency. It has been successfully used for genome editing in various cells and laboratory animals, including human cell lines, bacteria, zebrafish, yeast, mice, fruit flies, roundworms, pigs, and monkeys [29,30]. Plus, it is particularly important to select reasonable cell models for exploring viral replication. The choice of the cellular model system for each virus was, therefore, made based on the capacity of the cell lines to productively replicate each virus [31]. U251 cell lines, derived from a patient with glioblastoma multiforme, have been extensively used to study various aspects of viral infections [32]. Their ability to support the replication of different viruses makes them an excellent in vitro model for investigating viral pathogenesis and understanding the molecular mechanisms underlying viral life cycles. JEV can infect various cell lines, unlike DENV or ZIKV. Additionally, JEV is much more neuroinvasive and JEV invasion into the brain usually results in severe encephalitis [33]. Hence, U251 cells are commonly used as a model for studying the infection mechanisms of JEV. Thus, we also chose the U251 cell line to establish an infection model for studying JEV, which primarily infects neural cells. This selection validates the appropriateness of our research methodology.

In this study, HuR-sgRNA was designed and cloned into the LentiCRISPR V2 CRISPR/Cas9 plasmid for transfection into U251 cell lines. This genome editing approach differs from most prior studies manipulating HuR expression, which typically employ RNA interference or small molecule inhibitors to suppress HuR levels transiently [34,35,36]. Liang et al. knocked down HuR levels in HaCaT cells using siRNA and studied its regulatory effect on SQSTM1 mRNA expression [37]. Kim et al. performed a knockdown of the HuR level in OVCA433 and OVCAR10 cells and studied the regulatory mechanism of HuR on SOD2 expression [38]. Some studies also used inhibitors to suppress HuR. Zhang et al. utilized cycloheximide to inhibit HuR protein levels and explored the function of ATF3 [39]. Chellappan et al. employed SRI-42127 to inhibit the nuclear shuttling function of HuR, thus suppressing its protein activity [40]. Additionally, some studies obtained *HuR* gene knockout cells from mice with conditional *HuR* gene deletion for subsequent experiments [41,42]. We leveraged the CRISPR/Cas9 gene-editing system to entirely and permanently ablate endogenous HuR expression. HuR knockout was used to further verify the functions of the target gene, achieving higher knockout efficiency than the knockdown efficiency reported in some previous studies. This further eliminated the interference of residual HuR. These results provided a solid foundation for follow-up studies.

Current studies have shown that HuR is involved in the life process of the *flavivirus* and *alphavirus* infectious process. Still, it is important to note that different viruses may utilize HuR differently [8,16,43]. JEV belongs to the Flaviviridae family. We utilized JEV for follow-up experiments. *HuR* gene-knockout cells were infected with JEV. Then, HuR mRNA and protein expressions were determined using qRT-PCR and Western blot, respectively. We found that the replication of JEV was reduced in *HuR* gene knockout cells. This finding is consistent with the results of previous genome-wide CRISPR screens. This study has laid a solid experimental foundation for us, demonstrating the feasibility of utilizing CRISPR technology for *HuR* gene knockout and providing a unique platform for unravelling the complex interplay between HuR and viruses. These findings open up new avenues for understanding the molecular mechanisms underlying viral infections. However, further experiments are still required to strengthen the basis of our findings, such as establishing wild-type U251 cells as sgRNA controls and performing HuR overexpression and complementation experiments to investigate the effects on JEV replication. We hope we can consolidate this discovery through these additional experiments.

Our study demonstrated that knockout HuR can inhibit JEV replication, this result is the same as that reached by Harris et al. [44], Korf et al. [45], and Shwetha et al.’s [46] research results; they all found that HuR can significantly inhibit the levels of HCV replicon. Both JEV and HCV are members of the *Flaviviridae* family. But, JEV is a single-stranded RNA flavivirus transmitted by mosquitoes of the genus *Culex* [47]. JEV is an important zoonotic disease. In human beings, JEV infections can lead to mild febrile illness, while severe cases progress to fatal encephalitis with an approximate mortality rate of 30% [48]. Globally, approximately 68,000 cases of Japanese encephalitis virus infection occur annually, with children under 14 years old accounting for 75% of the total number. In animals, pigs infected with JEV usually suffer from reproductive disorders causing huge economic losses to the pig industry [49,50]. And, no effective antiviral drugs for JEV are currently available [51,52]. Hence, elucidating the host factors involved in JEV pathogenesis has crucial significance. Previously, through CRISPR/Cas9 gene-editing technology, we have identified HuR as a potential host factor influencing JEV infection. However, the role of HuR in JEV infection has not been reported. Therefore, further exploration of the function of HuR in JEV infection is of great significance. Our results represent an initial application of the *HuR* gene knockout cells to study viral replication rather than a mechanistic delineation of how HuR impacts JEV. However, our preliminary results indicate it would be meaningful to further explore the relationship between HuR and JEV in future studies. Our current findings demonstrate successful CRISPR/Cas9-mediated knockout of *HuR* gene in U251 cell lines, establishing a valuable tool for future investigation of how HuR inhibits JEV replication.

## 5. Conclusions

For the first time, knockout of the *HuR* gene in U251 cells was performed and confirmed by the CRISPR/Cas9 method. As a result of knocking out *HuR* gene, followed by a decrease in its mRNA and protein, JEV replication was inhibited, which was also observed in this study. Therefore, it is worthy of thorough exploring and investigation. This study lays a foundation for further exploration of the specific mechanism by which HuR influences the course of JEV infection and may contribute to developing novel therapeutic strategies targeting HuR-mediated viral processes.

## Figures and Tables

**Figure 1 microorganisms-12-00314-f001:**
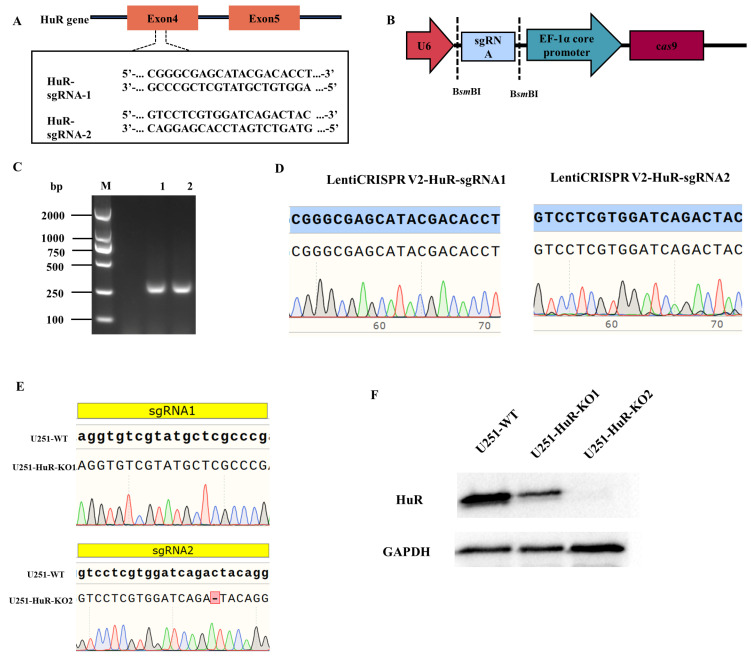
CRISPR/Cas9 gene editing technology knockedout the *HuR* gene in U251 cell lines. (**A**) The sgRNA sequences targeted exon 4 of HuR. (**B**) Schematic illustration of LentiCRISPR V2 plasmid with two restriction sites (BsmBI) existing in the target gene. (**C**) The sgRNA sequences were cloned into the LentiCRISPR V2 plasmid. LentiCRISPR V2 plasmid PCR products were analyzed by agarose gel electrophoresis. Lane1: LentiCRISPR V2-HuR-sgRNA1, Lane2: LentiCRISPR V2-HuR-sgRNA2. (**D**) Sanger sequencing confirmed successful ligation of sgRNA oligos into the LentiCRISPR V2-HuR-sgRNA1 and LentiCRISPR V2-HuR-sgRNA2 plasmids. (**E**) Using a limited dilution culture, we obtained single cell-derived clones from the U251 cell lines. Sanger sequencing results showed C nucleotide deletion in the U251-HuR-KO2 clone. (**F**) The HuR expression level was verified by Western blot analysis. Note: the green, red, blue and black colourful lines correspond to the A, T, C and G nucleobase signals, respectively.

**Figure 2 microorganisms-12-00314-f002:**
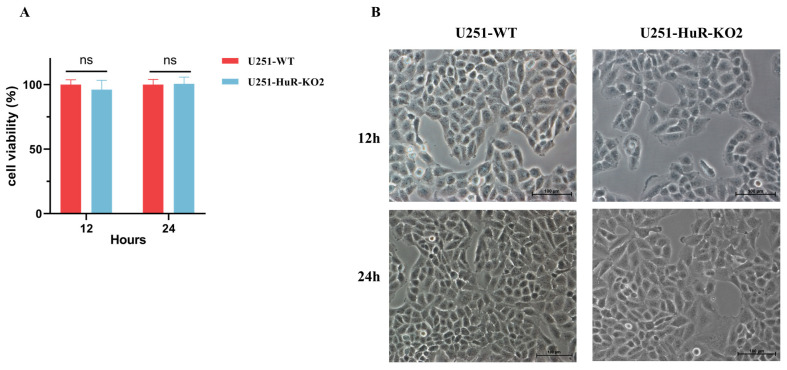
Assessing Cell Viability and Morphology using CCK-8 Assay and Light Microscopy. (**A**) To assess the cell growth, U251 cell lines and U251-HuR-KO2 cells were plated onto 96-well plates and cell viability was determined using CCK-8 assay (*n* = 5 experimental replicates, ns means *p* > 0.005). (**B**) Typical morphology of U251 cell lines and U251-HuR-KO2 clone cultured in DMEM medium for 12 h and 24 h (200×).

**Figure 3 microorganisms-12-00314-f003:**
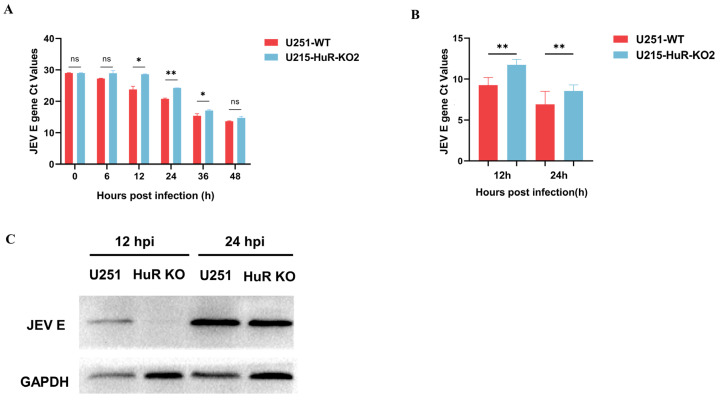
The impact of *HuR* gene knockout on JEV replication. (**A**) Supernatants were collected at various time points from JEV-infected U251-WT and U251-HuR-KO2 cells and viral loads quantified by qRT-PCR to get the JEV *E* gene Ct values. (**B**) U251-WT and U251-HuR- KO2 cells were infected by JEV after 12 h and 24 h post-infection (hpi), and quantified by qRT-PCR to get the JEV *E* gene Ct values. (**C**) The Western blot results for the U251-WT and U251-HuR-KO2 cells after infection by JEV 12 and 24 hpi. (* means *p* < 0.05, ** means *p* < 0.01, ns means *p* > 0.05).

**Table 1 microorganisms-12-00314-t001:** Primer sequences for sgRNA.

Target Names	Primer Sequences
HuR-sgRNA1	F: CACCGCGGGCGAGCATACGACACCT
	R: AAACAGGTGTCGTATGCTCGCCCGC
HuR-sgRNA2	F: CACCGTCCTCGTGGATCAGACTAC
	R: AAACGTAGTCTGATCCACGAGGAC

**Table 2 microorganisms-12-00314-t002:** Primer sequences for qRT-PCR.

Target Names	Primer Sequences
GAPDH-RT	F: AGAAGGCTGGGGCTCATTTG
	R: GGGGCCATCCACAGTCTTC
JEV-E-RT	F: TGGAGCCACTTGGGTG
	R: TGGAGCCACTTGGGTG

## Data Availability

The novel findings and contributions of this research are detailed in the published manuscript. Further questions regarding the study can be addressed to the authors’ listed correspondence.

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
