# Peer review of "CRISPR/Cas9-Mediated Knockout of the HuR Gene in U251 Cell Inhibits Japanese Encephalitis Virus Replication"

_microorganisms, 2024, doi:10.3390/microorganisms12020314_

Round 1

Reviewer 1 Report

Comments and Suggestions for Authors

The authors have good material and results. The manuscript needs a few refinements. Especially in the introduction and discussion section, the order of the information (paragraphs) could be reviewed. Particularly, for experimental studies the first paragraph of the Discussion section could be a summary of the results; it helps the reader to understand all the analyses performed.

I also think it is valid for the authors to add a paragraph on the perspectives and limitations of their study. I would also like to ask about the possibility of performing the knockout in other cell types, especially primary cultures from patients' cells.

Another suggestion is to review the style of the references in the Discussion section, which is different from the rest of the manuscript.

Minor: please italicize all the genes mentioned and the genus Culex (line 242).

Reviewer 2 Report

Comments and Suggestions for Authors

Dear editor, 

First of all, my apologies for the late reply, it's due to technical reasons. 

The manuscript written by Dr. Sai-Qi Luo et al., provides an interesting results and technique in studying and analysing the role of specific genes in the replication of viruses and virus-host interaction. 

I was interested in reading and reviewing the paper, yet in my opinion it still needs some improvement and work to be done. 

To make it easier for the authors and you, I uploaded a .pdf file where all my comments are added and marked in yellow. 

I hope the authors take it into consideration.

Regards, 

Round 2

Reviewer 2 Report

Comments and Suggestions for Authors

Dear editor,

I would like to thank the authors for sharing their work, answering my comments, and taking my suggestions into consideration. 

Up to me, I dont have anymore questions to the authors, and would recommend the paper for publication.

Regards,